# Unraveling the disease pyramid: the role of environmental micro-eukaryotes in amphibian resistance to the deadly fungal pathogen *Batrachochytrium dendrobatidis*

Rayan Bouchali,[1] Hugo Sentenac,[1,2] Kieran A. Bates,[3] Matthew C. Fisher,[4] Dirk S. Schmeller,[1] Adeline Loyau[1]

**ABSTRACT**   The disease pyramid conceptualizes the predictors of host infection risk, linking the host, the pathogen, environmental conditions, and both host and environmental microbiomes. However, the importance of the interaction between environmental and host-associated microbiomes in shaping infectious disease dynamics remains poorly understood. While the majority of studies have focused on bacteria, the role of micro-eukaryotes has been seldom investigated. Here, we explore three axes of the disease pyramid using an 18S rRNA gene metabarcoding approach to analyze the micro-eukaryotic assemblages of biofilm, water, and skin samples from three European amphibian species. Skin bacterial communities of the investigated amphibian populations have already been shown to be impacted by the presence of the lethal fungal pathogen *Batrachochytrium dendrobatidis* (*Bd*), with a higher abundance of protective bacteria in infected populations and a greater environmental microbial contribution to the skin microbiota in *Bd*-positive lakes. Here, we explored the relationships between the micro-eukaryotic skin communities of these tadpole populations with their surrounding environment. Tadpoles were sampled at 22 mountain lakes located in the Pyrenees (France), 8 of which harbored amphibian populations infected by *Bd*. We found that biofilms from *Bd*-negative lakes had higher environmental micro-eukaryotic diversity and a greater abundance of putative anti-*Bd* fungi, both in the environment and on the skin microbiota of *Bufo spinosus* and *Rana temporaria*, but not of *Alytes obstetricans*. Bayesian SourceTracker analysis further showed that the environmental contribution from biofilms to amphibian skin micro-eukaryotic assemblages was higher in *Bd*-positive lakes for *B. spinosus* and *R. temporaria*, but not for *A. obstetricans*.

**IMPORTANCE**  Research on host-associated microbiomes and infectious diseases has mostly focused on bacteria, overlooking the potential contributions of micro-eukaryotes to infection dynamics. Here, we show that environmental and skin-associated micro-eukaryotes—especially putative anti-*Batrachochytrium dendrobatidis* (*Bd*) fungi—differ between *Bd*-positive and *Bd*-negative amphibian populations in mountain lakes. Our results suggest that micro-eukaryotes influence disease resistance and microbiome assembly, similarly to bacteria. Importantly, environmental reservoirs of micro-eukaryotes appear to contribute differently across infection contexts. These findings demonstrate the importance of adopting a broader microbiome perspective that includes micro-eukaryotes when investigating the ecological mechanisms underlying infectious disease risk.

**KEYWORDS**   high mountain lakes, disease pyramid, microbial ecology, disease ecology, amphibian immunity, 18S rRNA gene, chytrid fungus, Bayesian models

Address correspondence to Rayan Bouchali, rayan.bouchali@toulouse-inp.fr.

Rayan Bouchali and Hugo Sentenac contributed equally to this article. The author order was determined based on a randomized selection process.

Dirk S. Schmeller and Adeline Loyau contributed equally to this article.

The authors declare no conflict of interest.

See the funding table on p. 14.

Since the role of the host-associated microbiome in disease dynamics has been recognized (1–3), its contribution to host immunity has become increasingly evident (4–6). In humans, exposure to less diverse environmental microbiomes is associated with reduced immune effectiveness, underscoring the vital role of environmental microbes (7). Yet, the role of environmental microbiomes remains little explored in natural settings, despite their potential importance both as a biological barrier limiting pathogen spread in the environment and as a reservoir of beneficial microbes that, once colonizing the host, enhance immunity and resilience (8, 9). This knowledge gap has led to the concept of the disease pyramid, an extension of the classical disease triangle (including host, pathogen, and environment). In the disease pyramid, host- and environment-associated microbiomes are considered as an additional dimension influencing disease outcome. The term "pyramid" highlights this expansion from three (triangle disease) to four interacting components, rather than implying a hierarchical organization, and under-scores their combined role at both individual and population levels (10–13). It has indeed been well established that populations are much less prone to outbreaks of infectious diseases when the ecosystem harbors micro-organisms that can inhibit the pathogen directly in the environment (14) and/or hamper the colonization by the pathogen through direct competition (15) or predation (16). The environmental microbiome can also promote host immunity by transferring communities of protective microbes to the host through coalescence processes (9).

Amphibians are one of the most well-studied vertebrate models for unraveling interactions between hosts, pathogens, the environment, and both host and environ-mental microbiomes. Interest in amphibians and their skin microbiome has emerged in the face of the global decline of this vertebrate group due to the panzootic chytridiomy-cosis, a skin disease caused by the fungal pathogen *Batrachochytrium dendrobatidis* (*Bd*) (17, 18). The amphibian skin microbiome is well known to contribute to host immunity by harboring protective bacterial communities (1, 19). For example, bacterial commun-ity structure and the relative abundance of protective bacteria have been shown to differ based on *Bd* exposure in amphibian populations (8), and some of the protective bacteria were found to come from the surrounding environment, such as biofilm, water, or sediments (5, 20, 21). Water was shown to contribute more to the amphibian skin bacterial communities in comparison to biofilm, and this contribution was higher in populations infected by *Bd* (R. Bouchali, H. Sentenac, D. S. Schmeller, A. Bernardo-Cravo, and A. Loyau, unpublished data).

Most prior studies seeking to unravel the interactions depicted in the disease pyramid have focused on bacteria only. In contrast, micro-eukaryotes, comprising fungi, protists (including apicomplexans), and other small microscopic eukaryotic organisms, have been largely overlooked and remain the least studied microbial paraphyletic group, with only 5% of metabarcoding projects devoted to their diversity and ecology (22). While representing only 1%–5% of total microbial cells, the functional roles of micro-eukar-yotes should not be underestimated, as they are important metabolite producers (23) and typically have much larger cell sizes than prokaryotes, enabling single organisms to occupy relatively large surface areas. Micro-eukaryotes also play key roles in cross-king-dom interactions. For example, certain taxa promote the diversity of favorable bacteria and, in the human gut, enhance microbiome diversity by preventing the colonization of dominant taxa (24). At the microscale, food webs involve predation among micro-eukar-yotes as well as on bacteria, processes that can shape microbial community dynamics and composition across all microbial kingdoms (25). More specifically, yeasts may also exhibit anti-toxin properties against pathogenic bacteria such as *Vibrio cholerae* and harbor anti-bacterial and anti-viral functions that are important in shaping the microbial communities of the human gut (23). Micro-eukaryotes may also interact directly with host immunity; for example, the protozoan *Tritrichomonas musculis* can decrease the risk of bacterial infection in a vertebrate host by activating the inflammasome (26).

Studies on the micro-eukaryotic communities inhabiting amphibian skin are very scarce, mainly focusing on fungal communities (27). In amphibians, fungi are indeed

the most abundant micro-eukaryotes, making up 60% of total eukaryotic cells (28). Fungi have also been found to be highly efficient in inhibiting *Bd*, with almost half of the amphibian skin mycobiome showing anti-*Bd* properties *in vitro* (29), including the isolates belonging to the *Neobulgaria* and *Pleosporales* genera (30). However, fungal communities colonizing amphibian skin did not differ according to *Bd* infection status, but infection was correlated to specific fungal operational taxonomic units (31). While environmental micro-eukaryotic communities can directly influence pathogen survival, host microbiota, and thus host health, their broader function within the disease pyramid is not understood. Some micro-eukaryotic organisms have shown anti-*Bd* properties, especially ciliates, with one study demonstrating a reduction in *Bd* zoospore abundance driven by predatory ciliates, such as *Paramecium* and *Spirostomum* (16). Ciliates residing, even temporarily, on the skin may also be key players in colonization resistance against *Bd*. Many larval amphibians also extensively feed on environmental biofilms (32), which are crucial for aquatic ecosystems and harbor many micro-eukaryotes (33). The latter might positively influence host health through multiple ways, which remain largely unknown: by containing vital nutrients (34), by inactivating/consuming *Bd* motile zoospores (14), and by contributing to the enrichment of the skin and gut host microbiome through coalescence processes.

Here, we used 18S rRNA metabarcoding and the latest microbiome analysis tools to investigate the roles of both environmental and host-associated micro-eukaryotes in driving resistance to Bd in high mountain lakes of the Pyrenees. First, we assessed the importance of environmental micro-eukaryotic assemblages in host protection against *Bd* in natural populations of three amphibian species. Second, we infer functional anti-*Bd* abilities in amphibian skin, biofilm, and water (planktonic) samples. Specifically, we tested the following hypotheses: (i) the environmental microbiome acts as a biological barrier against *Bd*, reflected by a greater prevalence and abundance of putative anti-*Bd* microorganisms in pathogen-free lakes, and (ii) there is a close relationship between the amphibian skin and the environmental microbiome through transfers and engraftment of putative anti-*Bd* micro-eukaryotic cells.

## RESULTS

### Micro-eukaryotic communities of amphibian tadpole skin and their living environment

#### Micro-eukaryotic diversity and richness

For all compartments, we did not detect an effect of sampling events (i.e., season and year) on the α-diversity (ANOVA, all $P > 0.01$). Biofilm samples showed the highest value for Shannon and Simpson indexes, indicating a higher diversity with both dominant and rare species, as well as for amplicon sequence variant (ASV) richness (ANOVA, all $P < 0.01$; Fig. S1; Tables S1 and S2). Skin micro-eukaryotic communities had a low diversity and richness (richness: from 11 to 348 ASVs; Shannon: 0.37–4.80; Simpson: 0.13–0.98; evenness: 0.1–1), with significant dissimilarities between amphibian species (Fig. S1; Tables S1 and S2).

The β-diversity was significantly different in the ASV profiles of biofilm and water samples (PERMANOVA, $R^2 = 0.59$, $P < 0.01$; Fig. S2), as well as in regard to the lake of origin (PERMANOVA, $R^2 = 0.86$, $P < 0.01$). The β-diversity of amphibian skin was not different from either water or biofilm microbiomes (biofilm: PERMANOVA: $R^2 < 0.01$, $P = 0.43$; water: PERMANOVA: $R^2 = -0.02$, $P = 0.98$). It was determined by the lake of origin (PERMANOVA, $R^2 = 0.21$, $P < 0.01$) and not species-specific (*Envfit* model adjusted with the lake effect: $R^2 = 0.01$, $P > 0.05$).

We compared the richness and diversity of micro-eukaryotic (18S ASVs) and bacterial (16S ASVs) skin communities from the same amphibian population, using 16S data from a previous study (8). Only ASV richness (linear regression, all $P < 0.05$) was correlated between bacterial and micro-eukaryotic communities (Fig. S3). For biofilm and water, we found a significant congruence between the ordinations of the bacterial

and micro-eukaryotic communities (protest, $m^2 = 0.16$, $r = 0.92$, $P = 0.001$) (Fig. S4a). For amphibian skin communities, significant congruence between 16S and 18S ordinations was detected in all three amphibian species (*Alytes obstetricans*: $m^2 = 0.83$, $r = 0.41$, $P = 0.011$; *Bufo spinosus*: $m^2 = 0.15$, $r = 0.92$, $P = 0.001$; *R. temporaria*: $m^2 = 0.66$, $r = 0.59$, $P = 0.001$, Fig. S4b).

## Micro-eukaryotic and protective taxa inhabiting the amphibian tadpole skin and their environment

We characterized the most abundant micro-eukaryotic taxa in amphibian skin, water, and biofilm samples (Table S3). In all amphibian species, *Vorticella* dominated skin microbiomes, but we also observed species-specific enrichment of genera (ANCOM-BC, all $P < 0.05$, Fig. S5). Dominance of ASVs was also different between the amphibian species, biofilm, and water (Tables S4 and S5; Fig. S5). The five most abundant genera in biofilms were generally rare in water (<0.5%), and conversely, the five most abundant genera in water were rare in biofilms. Characterization of micro-eukaryotic taxa in amphibian skin, biofilm, and water samples allowed detection of four putative anti-*Bd* fungal genera: *Aspergillus*, *Basidiobolus*, *Cyberlindnera*, and *Penicillium*. Only *Aspergillus* and *Penicillium* were detected on skin (*A. obstetricans*: 0.07% and 0.41%; *B. spinosus*: 0.08% and 2.33%; and *R. temporaria*: 0.09% and 0.29%), with no significant differences between species (ANOVA, $P > 0.05$; Fig. 1; Table S6). In biofilm and water, the four genera were rare (*Penicillium* 0.01% and 0.02%; *Basidiobolus* 0.01% and <0.01%; *Aspergillus* <0.01% and 0.00%; *Cyberlindnera* < 0.01%; ANOVA, $P > 0.05$; Table S6). *Ciliata* and *Rotifera* were abundant on tadpole skin and in the environment, with higher *Rotifera* proportions in *A. obstetricans* and water, and higher *Ciliata* proportions in *B. spinosus* and biofilms (ANOVA, $P < 0.01$; Table S6). Five *Ciliata* genera harboring anti-*Bd* species (*Blepharisma*, *Dileptus*, *Euplotes*, *Paramecium*, and *Spirostomum*) showed comparable abundances across hosts, water, and biofilms (ANOVA, $P > 0.05$; Table S6).

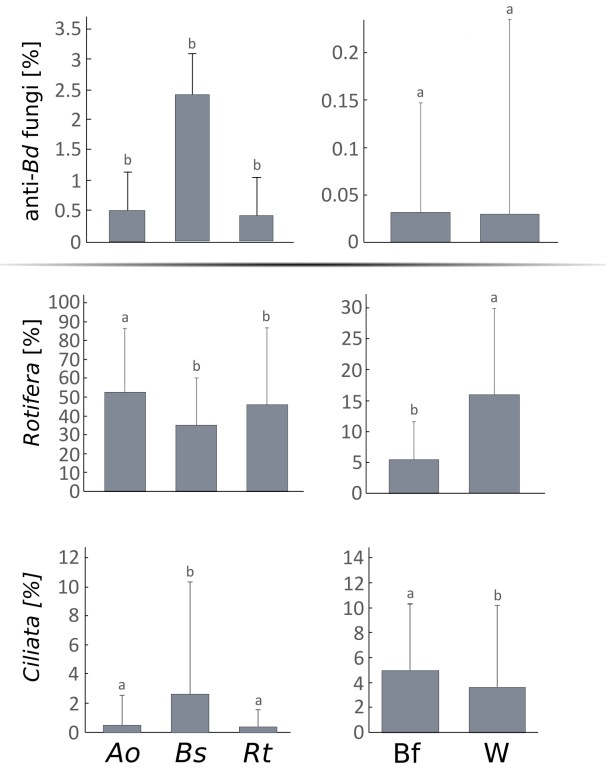

**FIG 1** Relative abundance of anti-*Bd* fungi, *Ciliata,* and *Rotifera* in samples from the skin of three amphibian species, environmental biofilm, and water. The different letters indicate statistical groups that are significantly different from each other (ANOVA test followed by Tukey's *post hoc* test, $P < 0.05$).

## Micro-eukaryotic assemblage and *Bd* infections

### Relationships between Bd and the micro-eukaryotic diversity and richness

The α-diversity of tadpole skin and biofilm, but not water, differed significantly between *Bd*-positive and *Bd*-negative lakes (Fig. 2; Fig. S6 and S7; Tables S1, S2, and S7). ASV richness was lower in *Bd*-positive lakes for *A. obstetricans*, *B. spinosus*, and *R. temporaria* (Wilcoxon test, all $P < 0.01$) (Fig. S6; Tables S2 and S7), whereas Shannon and Simpson diversity on skin did not differ (Wilcoxon test, all $P > 0.05$) (Table S7). In biofilms, micro-eukaryotic diversity and evenness were higher in *Bd*-negative lakes (Shannon: $W_{46} = 4,116$, $P < 0.01$; Simpson: $W_{15} = 3,969$, $P < 0.01$; evenness: $W_{34} = 3,727$, $P < 0.05$), while ASV richness showed no significant difference ($W_{50} = 3,635$, $P > 0.05$). No differences with regard to the infection status of the amphibian population were observed in water micro-eukaryotic communities (all $P > 0.05$; Fig. S6; Table S7).

The β-diversity of micro-eukaryotic assemblages was not different between amphibian skin samples from *Bd*-positive and *Bd*-negative lakes (PERMANOVA, $R^2 = 0.01$, $P = 0.33$), but significant differences were apparent in biofilm (PERMANOVA, $R^2 = 0.15$, $P < 0.01$) and water community compositions (PERMANOVA, $R^2 = 0.08$, $P < 0.01$) (Fig. S2).

### Relation between Bd presence and the abundance of protective micro-eukaryotic taxa

Genera differentially distributed according to *Bd* infection status were identified for all compartments using ANCOM-BC (Fig. 3). None of the genera showing differentiated repartitions were putative anti-*Bd* taxa; however, the total abundance of putative anti-*Bd* micro-eukaryotes was associated with *Bd* presence (Fig. 2 and 4). In *B. spinosus* and *R. temporaria*, but not *A. obstetricans*, putative anti-*Bd* fungi were more abundant on skin from *Bd*-negative lakes (5.82%, 0.47%, and 0.59%, respectively) than *Bd*-positive lakes (0.00%, 0.01%, and 0.36%, respectively) (Wilcoxon test, $W_9 = 45$, $P < 0.05$; $W_{113} = 2$, $P < 0.05$; and $W_{113} = 0.45$, $P > 0.05$; Fig. 2 and 4A; Table S8). Ciliata were more abundant on amphibian skin from *Bd*-positive lakes across all species (Wilcoxon test, all $P < 0.05$; Fig. 4C; Table S8), while *Rotifera* showed no differences (all $P > 0.05$; Fig. 4B). Biofilm and water samples had higher putative anti-*Bd* fungi in *Bd*-negative lakes, but *Bd* presence was not linked to *Rotifera* or *Ciliata* abundance (Wilcoxon test, all $P > 0.05$; Fig. 4; Table S8). Relative abundance of putative anti-*Bd Ciliata* genera did not differ with *Bd* infection,

| | *Bd* positive population | | | | |
| --- | --- | --- | --- | --- | --- |
| | *Ao* | *Bs* | *Rt* | Bf | W |
| ASV richness | ↓ | ↓ | ↓ | | |
| Shannon | | | | ↓ | |
| Simpson | | | | ↓ | |
| Evenness | ↑ | ↑ | ↑ | ↓ | |
| Anti-*Bd* fungi | | ↓ | ↓ | ↓ | ↓ |
| *Ciliata* | ↑ | ↑ | ↑ | | |
| *Rotifera* | | | | | |
| Water contribution | | | ↑ | | |
| Biofilm contribution | | ↑ | ↑ | | |

FIG 2 Impact of the presence of *Bd* on the micro-eukaryotic communities of amphibian skin, biofilm, and water compartments, showing the significant increase (green) or decrease (red) of the micro-eukaryotic parameter values. *Ao, A. obstetricans; Bs, B. spinosus; Rt, R. temporaria*; Bf, biofilm; and W, water.

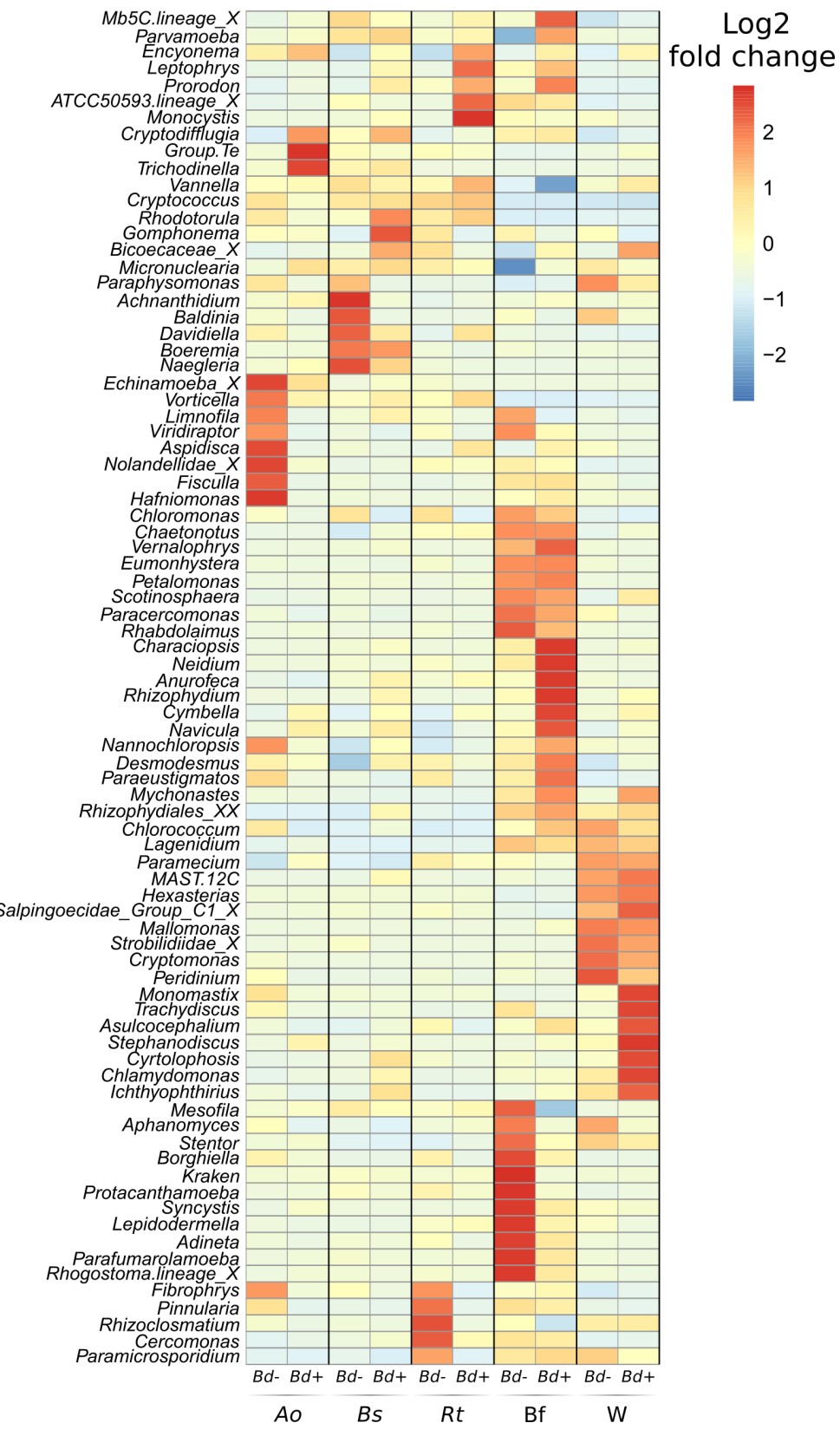

**FIG 3** Heatmap of the micro-eukaryotic genera differently associated with the host species, the biofilm, and the water samples, and according to the *Bd* infection status (negative vs positive), as revealed by the analysis of compositions of microbiomes with bias correction (ANCOM-BC). When the genus was unknown, the family (_X) or the order (_XX) is indicated. Only taxa with a significant overall distribution bias are shown.

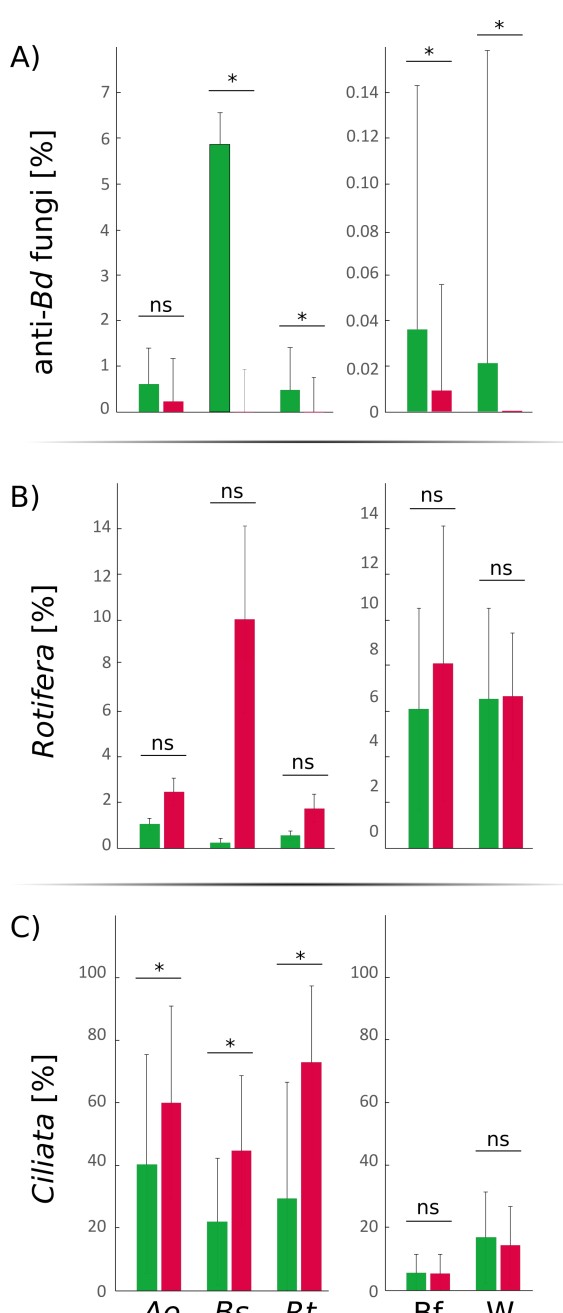

**FIG 4** Average relative abundance of the (A) putative anti-*Bd* fungi, (B) *Rotifera,* and (C) *Ciliata* according to the host species and the *Bd* infection status of the considered lakes (green, *Bd* negative; red, *Bd* positive). Asterisks highlight significant differentiation between *Bd*-positive and *Bd*-negative lakes (Wilcoxon test). *Ao, A. obstetricans; Bs B. spinosus, Rt, R. temporaria*; Bf, biofilm; and W, water.

except for *A. obstetricans*, which was higher in *Bd*-positive lakes ($W_7 = 3,570$, $P > 0.05$; Table S8).

### Coalescence processes linked to Bd infection status

The SourceTracker analysis revealed that environmental biofilm and water communities contributed significantly to the skin micro-eukaryotic communities of *A. obstetricans* (14 % ± 19% and 14% ± 21%), *B. spinosus* (20% ± 16% and 16% ± 20%), and *R. temporaria* (19% ± 21% and 19% ± 30%) (Tables S9 and S10). *Bd* infection status influenced

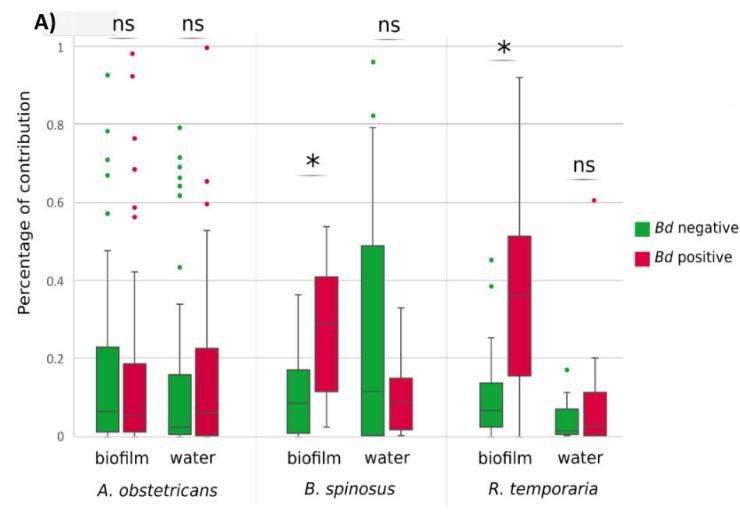

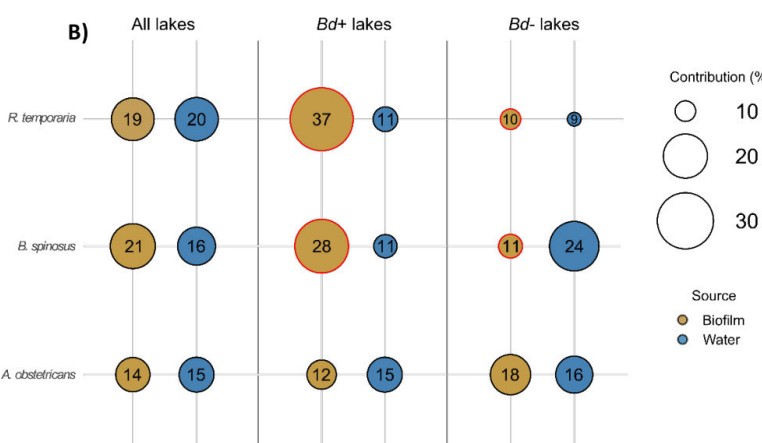

**FIG 5** (A) Boxplots showing the inferred percentage of contribution from biofilm and water samples to the assembly of amphibian skin micro-eukaryotic communities, as estimated by Bayesian SourceTracker analysis. (B) Mean inferred environmental contributions to amphibian skin communities. Asterisks and red circles denote significant differences in contributions between *Bd*-positive and *Bd*-negative lakes (Wilcoxon test).

these environmental transfers (Fig. 2 and 5; Tables S9 and S11). Biofilm contributions were higher in *Bd*-positive lakes than *Bd*-negative lakes for *R. temporaria* (37% ± 25% vs 9% ± 10%; $W_{20} = 130$, $P < 0.01$) and *B. spinosus* (27% ± 15% vs 10% ± 10%; $W_9 = 26$, $P < 0.01$), but not for *A. obstetricans* (12% ± 16% vs 18% ± 23%; $W_{67} = 4{,}112$, $P > 0.05$). Water contributions to skin microbiota did not differ with Bd infection status (Wilcoxon, all $P > 0.05$; Fig. 5; Table S11).

A phylogenetic null model showed that micro-eukaryotic communities of amphibian skin are driven mostly by stochastic processes, with low average absolute βNTI values (<1) and high variability between individuals. The presence of *Bd* did not affect the assembly processes, with similar βNTI values between amphibian skin from *Bd*-positive and *Bd*-negative lakes (ANOVA, $F_7 = 1.422$, $P = 0.20$).

## DISCUSSION

Our study examined the links between micro-eukaryotic communities on tadpole skin and environmental sources, as well as their relationship to the presence of the fungal amphibian pathogen *Bd*. Our study builds on recent experimental evidence showing that environmental biofilms can inactivate *Bd* zoospores (14), suggesting that the

environmental microbiome could prevent the invasion of ecosystems by the pathogen. Our metabarcoding approach provides ecological context to these mechanistic findings, integrating amphibian skin, biofilm, and water microbiomes. While metabarcoding alone does not establish causality, it allows us to situate field-based observations within the framework of the disease pyramid (11) by identifying associations between microbial assemblages and *Bd* occurrence. Our results show that environmental and amphibian skin micro-eukaryotic communities differed with the *Bd* infection status of the studied amphibian populations. Biofilm α-diversity, but not water α-diversity, was reduced in infected lakes, suggesting that the lower diversity of micro-eukaryotic communities in biofilms from infected lakes may have favored the environmental invasion by *Bd*.

The biotic resistance theory suggests that a rich biodiversity and its associated interactions (e.g., competition and predation) could prevent the invasion of undesirable species (35). Here, rich and diverse biofilms may be more stable and more resistant to perturbation, increasing the probability of inactivation of *Bd* zoospores, the infectious stage of *Bd* (i.e., biological barrier) (14, 36). The dominance of some species (i.e., low evenness) can lead to a decline in microbial interactions and metabolic and mechanistic functions that are crucial for inhibiting pathogen establishment in the environment (37). In biofilms from *Bd*-positive lakes, the lower community diversity may reduce the inhibitory priority effect (38, 39), leaving ecological niches vacant and open to pathogen invasion (40). Conversely, in richer biofilms, as observed in *Bd*-negative lakes, established micro-eukaryotes may influence community assembly by occupying niches and using resources, thereby reducing the opportunities of establishment for newly arriving species, including emerging pathogens (41). A richer biofilm also increases the probability of containing protective micro-eukaryotes, including *Bd*-inhibitory ones, by increasing the diversity of functions (42). Our results also suggest that, in addition to excluding *Bd* indirectly by eliminating available niches, micro-eukaryotic communities may also be able to directly inhibit the pathogen in the environment and prevent its installation, for example, by the production of anti-fungal compounds or by predation.

On the tadpole skin, we did not observe higher abundances of anti-*Bd* fungi in *Bd*-positive lakes, but *Ciliata* were generally more abundant. *Ciliata* are unicellular microscopic predators that can feed on lipid-rich *Bd* zoospores (16), but they are generally less versatile compared to fungi (i.e., they are more specialized and less able to use a variety of nutrient sources). They can rapidly proliferate in response to a sudden increase in a specific resource (43) and may benefit from the presence of *Bd* on the skin of amphibian tadpoles, which represents a significant nutrient source in oligotrophic environments (44). Infection by *Bd* and the induced dysbiosis in the skin microbiome (5, 45) could create a new ecological niche that *Ciliata* can exploit particularly well, similar to opportunistic fast-growing bacteria (8). *Ciliata* are more mobile compared to fungi and, therefore, are able to reach and colonize such a new niche, in our case, likely from biofilms on which tadpoles graze. Conversely, fungi are more versatile organisms and more efficient in oligotrophic settings as found in most mountain lakes (46), where they can exploit a broader range of nutrient sources and persist in the absence of *Bd*. Anti-*Bd* fungi may therefore be maintained through other interactions, including predation on other environmental microbes or utilization of alternative carbon sources, thereby contributing to a more stable and protective fungal community. Such a community may potentially prevent the establishment of *Bd* by consuming or otherwise inactivating its zoospores during colonization attempts. In contrast, in *Bd*-positive lakes, the proliferation of *Bd* and the associated dysbiosis could create conditions less favorable for fungi but more suitable for opportunistic protists such as ciliates.

The differentiated distributions of anti-*Bd* fungi and *Ciliata* illustrate the difficulty of investigating the ecological interactions between the host, the host microbiome, *Bd,* and environmental microbial communities, as depicted in the disease pyramid. The temporality of two processes, first the colonization of the emerging pathogen prevented (or not) by the environmental micro-eukaryotic communities, and second, the response of the amphibian skin microbiome once *Bd* is established in amphibian assemblages,

is not easily disentangled. Amphibians can respond adaptively to *Bd* with their skin microbiome shifting over time, as suggested by the adaptive microbiome theory (5). A key question is whether this process favors the proliferation of *Ciliata* that are capable of predating upon *Bd*, thus enhancing protection against the pathogen. Such an increase in protective microorganisms can be followed by a decrease in richness and an increase in dominance of beneficial microorganisms (5). We indeed found a lower ASV richness on tadpole skin from *Bd*-positive lakes, which may reflect enrichment in protective micro-eukaryotes that hamper *Bd* infections. The impact of *Bd* on the enrichment of microbes from environmental sources was already observed for bacteria (R. Bouchali, H. Sentenac, D. S. Schmeller, A. Bernardo-Cravo, and A. Loyau, unpublished data). However, only planktonic aquatic bacteria were enriched within the amphibian skin tadpoles, whereas micro-eukaryotes from biofilms were enriched when *Bd* was present. Planktonic micro-eukaryotes are adapted to life in suspension (47). Their ability to adhere to surfaces is often weaker than that of organisms living in biofilms. Conversely, biofilm micro-eukaryotes are specialized in adhesion and growth on solid substrates. They often possess structures that facilitate their attachment to surfaces (e.g., filaments, cilia, and flagella). Links between *Bd* presence and bacterial or micro-eukaryotic transfers (from the environment to the amphibian skin) were only observed for *B. spinosus* and *R. temporaria* (R. Bouchali, H. Sentenac, D. S. Schmeller, A. Bernardo-Cravo, and A. Loyau, unpublished data), whereas the inferred contribution from environments to the skin microbiota of *A. obstetricans* was lower. Between these species, there is a marked difference in the life-history trait of brumation. *A. obstetricans* has overwintering tadpoles that can spend several winters in the lake before metamorphosis, while the other two species complete their cycle from egg to metamorph within the same season. The longer exposure of *A. obstetricans* to lake environmental microbial communities may create inhibitory effects that would not allow a rapid adaptation to changing pathogen exposure.

In a previous study, we showed an opposite response to *Bd* for bacteria colonizing amphibian skin, which originated more frequently from water (R. Bouchali, H. Sentenac, D. S. Schmeller, A. Bernardo-Cravo, and A. Loyau, unpublished data), while micro-eukaryotes may originate more frequently from biofilms. This suggests complementary mechanisms to shape the skin microbial communities that ultimately lead to a more or less effective protection against skin pathogens. Bacteria, micro-eukaryote communities, and other microbial organisms are interdependent (48), and some can recruit others, thus influencing their dynamics and ecological function (23), even within amphibian skin microbiota (49). Therefore, integrated approaches, based on holistic frameworks such as the disease pyramid, are needed to grasp the coupling between the structures and functions of microbial communities. These approaches would enable a deeper understanding of the diversity of interactions occurring in ecosystems and their roles in warranting resilience in the face of increasingly numerous perturbations. Such knowledge could then be leveraged to develop more general forms of ecosystem protection, built on nature-based solutions and reinforcement of ecosystem resilience.

## MATERIALS AND METHODS

### Study sites and sampling strategy

We sampled tadpole populations of three amphibian species, *Alytes obstetricans*, *Bufo spinosus*, and *Rana temporaria*, inhabiting 22 high mountain lakes (between 1,063 and 2,522 m above sea level) located in the French and Spanish Pyrenean Mountains (Europe) (Table S12). Our lake system covers an area spreading over 173 km from East to West and 16 km from South to North (2,768 km$^2$). We focused on the tadpole skin microbiome because the larval stage is the point at which infection is established. Although tadpole skin is not fully keratinized and *Bd* primarily colonizes their keratinized mouthpart without lethality at this stage (50, 51), tadpoles can act as a reservoir that maintains and amplifies the pathogen in aquatic environments until metamorphosis, when susceptibility and mortality dramatically increase. Furthermore, the tadpole skin microbiome is of

particular importance, as it seeds the metamorphic microbiome (52), which becomes critical when *Bd* shifts from colonizing mouthparts to keratinized skin during and after metamorphosis (50, 53, 54). Hence, the tadpole skin microbiome is of vital importance for resistance to infection later in development due to vertical transfer from tadpoles over metamorphs to adults (50, 53).

Five to ten tadpoles of each species were captured in the littoral zone close to the shore of each lake by dip netting. To investigate skin micro-eukaryotic communities, tadpoles were swabbed across their full body using a sterile dry swab (MW100, MWE Medical Wire, Corsham, UK). Mountain lakes were visited several times during early, mid, or late summer, two times in 2016, and three times in 2017 and 2018. In the same lakes, tadpoles were monitored for over 10 years for the presence of *Bd* (30 tadpoles were captured and sampled at each instance when present), and a qPCR was run in duplicate (8). Eight lakes had amphibian populations infected for at least 10 years (*Bd*-positive lakes), 12 had populations negative for *Bd* (*Bd*-negative lakes), and 2 had a changing infection status (defined as "changing") (Table S12). Amphibian species showed different distributions: tadpoles of *A. obstetricans* were sampled across 13 lakes (in 2016, 2017, and 2018), *R. temporaria* across 15 lakes (only in 2017 and 2018), and *B. spinosus* across 3 lakes (only in 2018). This led to a total of 552 amphibian tadpoles swabbed, including 336 *A. obstetricans*, 164 *R. temporaria,* and 25 *B. spinosus*. Additionally, water samples were collected in the littoral zone where tadpoles live, using a clean Nalgene bottle disinfected with chlorhexidine and rinsed with physiological serum (NaCl 0.9%) and then lake water. Between 250 mL and 1 L of water, depending on the suspended matter in a given lake, were filtered with a 0.22 µm filter using a Nalgene filter holder and a vacuum hand pump. Biofilm was sampled by scraping rocks that were found at a depth of 15–30 cm, using a metal spatula that was disinfected with chlorhexidine and then rinsed with sterile physiological serum (NaCl 0.9%), and then put in a sterile 2 mL Eppendorf tube. All samples (swabs, biofilms, and filters) were immediately frozen on dry ice (−78°C) in the field and then stored at −25°C until extraction, which was usually performed the following month. This led to a total of 179 water and 203 biofilm samples (Table S12).

## 18S rRNA gene PCR amplification, sequencing, and metabarcoding pipeline

DNA from swabs was extracted with the Macherey-Nagel NucleoSpin Soil Kit (Valencia, CA, USA) according to the manufacturer's protocol. Negative controls included the 0.22 µm filter, extraction kit, and PCR reagents. The V8–V9 regions of the 16S rRNA gene were amplified using V8f and 1510r primers (55, 56), with a PCR program of 3 min at 95°C, 30 cycles of 30 s at 55°C and 30 s at 72°C, and a final 5-min extension at 72°C. Amplicon quality was checked on a 1.5% agarose gel. Illumina MiSeq sequencing of PCR amplicons (2 × 250 bp) was performed by the GENOTOUL platform (Toulouse, France). Primers, linkers, and barcodes were removed with Cutadapt version 4.0 (Python 3.9.12) (57). Raw reads were processed with DADA2 version 1.26.0 (58) following the paired-end SOP (https://benjjneb.github.io/dada2/bigdata_paired.html). Briefly, reads were filtered with minLen = c(200, 200), truncLen = c(280, 260), maxN = 0, maxEE = c (5, 5), and truncQ = 2. Chimeras were removed using the consensus method from removeBimeraDenovo. ASVs were assigned taxonomy with SILVA version 138.1 using the Wang classifier (minimum bootstrap 80%) (59). Contaminant ASVs were removed with Decontam (prevalence method, threshold 0.5) (60). ASVs affiliated to large multicellular plants, such as *Embryophyceae*, and metazoans, such as *Vertebrata*, *Arthropoda*, *Platyhelminthes*, *Annelida,* and *Mollusca*, were discarded. The Decontam package did not detect contaminant ASVs. Samples with less than 1,000 reads and/or 10 ASVs were discarded. This led to a final data set of 16,717,680 reads and 26,536 ASVs across 294 *A. obstetricans*, 139 *R. temporaria*, 24 *B. spinosus*, 188 biofilms, and 156 water samples (see Table S12). The taxonomic affiliation of the resulting ASVs allowed the classification of 97% of the reads at the class level, comprising 1,006 genera (80% assignment rate) and 1,189 species (68%).

## Statistical analyses

### Analysis of micro-eukaryotic richness and diversity

We first checked that not rarefying the micro-eukaryotic data set did not bias the results by comparing our unrarefied data with a data set rarefied to 1,010 reads. Rarefaction curves further indicated that sequencing depth was insufficient within the rarefied data set to capture the full micro-eukaryotic diversity, making rarefaction inappropriate (Fig. S7). Additionally, α-diversity indices were highly correlated ($R^2$ = 0.99) between the rarefied and unrarefied data set, and beta-diversity analyses (Bray–Curtis, PERMA-NOVA) showed no significant differences ($R^2$ < 0.0001, $P$ = 0.168), supporting the use of unrarefied data.

All the statistical analyses were computed using the R software version 4.3.2. Data distributions were checked using the *check_distribution* from the R package Performance. Rarefaction curves were computed with the vegan R package version 2.6.4 (61) to check the sequencing depth (61) (Fig. S9). Micro-eukaryotic diversity (Shannon, Simpson, and Pielou's evenness) and richness (ASVs number) indexes of amphibian skin and environmental samples, as well as β-diversity metrics (Bray–Curtis dissimilarity distances, PERMANOVA statistical tests, and NMDS), were computed with the vegan package. The chosen α-diversity indexes allowed a complementary insight into the microbial richness, with the Shannon diversity index being more sensitive to rare ASVs compared to the Simpson index, and Pielou's evenness describing the homogeneity of the ASV distributions in each sample. We used Bray–Curtis dissimilarity as our main metric because it accounts for both the presence of taxa and their relative abundances, which are key to assessing host–microbe interactions and resistance to Bd. This choice is particularly relevant since *Bd* infection may alter the abundance of resident taxa rather than introducing new ones (8). PERMANOVAs were run using the following predictor variables: amphibian host species, environmental matrices (i.e., biofilm and water), and their combination with the *Bd* infectious status. ANOVAs and Tukey's *post hoc* tests were used to compare the micro-eukaryotic richness and diversity indexes of amphibian skin samples according to the sampling season (early, mid, or late summer), as well as between amphibian species, and between biofilm and water. Due to a lower number of samples, we used Wilcoxon tests to compare the diversity indexes and richness between *Bd*-negative and *Bd*-positive lakes.

### Comparison of the bacterial and micro-eukaryotic community richness and diversity

Bacterial sequences (16S rRNA) were extracted from accession number PRJEB46609 (ERP130803) and processed as described previously (8). α-diversity of 16S and 18S was compared using linear regression with the *lm* function of the R software. Linear regressions were illustrated with the ggplot2 version 3.5.1 package (62). β-diversity (Bray–Curtis ordinations) of 16S and 18S was compared using the *procrustes* function of the Vegan package performed on the NMDS ordinations. Significance of the congruence was tested using the Vegan *protest* function (permutation test, $n$ = 999 permutations), a significant *P*-value meaning that there is a high similarity (=significant correlation) between the two microbial assemblages.

### Characterization of the abundance of putative Bd-inhibitory micro-eukaryotes

Putative anti-*Bd* genera were identified using the database from Kearns et al. (29), which includes 45 genera shown to mitigate the infection of amphibians by *Bd*. We also analyzed the abundance of the *Ciliata* and *Rotifera* phyla, which are likely to reduce the abundance of *Bd* by predation (16, 63). This also allowed the characterization of putative anti-*Bd Ciliata* and *Rotifera* genera, which harbor species showing anti-*Bd* properties under experimental conditions. We used ANOVAs followed by a Tukey's *post hoc* test to compare the abundance of these taxa between the different compartments and

Wilcoxon tests to compare *Bd*-negative and *Bd*-positive lakes. In addition, we tested the differential abundance of genus between amphibian skin, biofilm, and water, according to the *Bd* infectious status, using the ANCOM-BC R package (64) (function *ancombc2*, FDR method of *P*-value adjustment).

### Estimation of the micro-eukaryotic environmental contribution in the building of amphibian tadpole skin communities

The contribution of biofilm and water in the building of the micro-eukaryotic communities of amphibian skin was estimated using the Bayesian SourceTracker (65). SourceTracker was run with the following parameters: rarefaction=1,000 reads, burn-in=1,000, and restart=10.

A phylogenetic null model was used to analyze the forces that drive the micro-eukaryotic community assemblies, using the Beta Nearest Taxon index, a phylogenetic metric that measures how different two microbial communities are in terms of the evolutionary relatedness of their species (66). The Clustal Omega software was used to align the 18S rRNA gene ASV sequences (67). We used the MAFFT software to compute and build the Neighbor-Joining phylogenetic tree (68). The *cophenetic* function from the Stats base R package was used to estimate the phylogenetic distance between sequences. Mean Nearest Taxon Distances (MNTD) were computed with the function *mntd* of the Picante R package version 1.8.2, with the relative abundance of each ASV. MNTD null distribution was computed from random permutations ($n = 999$). βNTI values were computed using the formula:

$$\beta\text{NTI} = \frac{\text{MNTD}_{\text{obs}} - \text{MNTD}_{\text{rand}}}{\text{SD}(\text{MNTD}_{\text{rand}})},$$

with $\text{MNTD}_{\text{obs}}$ being the average distance to the nearest taxon observed in the community, $\text{MNTD}_{\text{rand}}$ is the average of the null MNTD distances, and $\text{SD}(\text{MNTD}_{\text{rand}})$ is the standard deviation of null *MNTD* distances. βNTI values near zero show a fully stochastic distribution of communities, whereas values farther from zero indicate the existence of deterministic processes.

### ACKNOWLEDGMENTS

We thank Judit Laufer and Adriana Bernardo-Cravo for performing some of the sequencing work of the water and skin microbiota. We also express our thanks to all student helpers, as well as to Pilar Durantez, Sylvain Lamothe, Frederic Julien, Oliver Machate, and others for support during and after fieldwork.

The project People, Pollution, and Pathogens (P[3]) was financed through the call "Mountains as Sentinels of Change" by the Belmont-Forum (ANR-15-MASC-0001-P3, DFG-SCHM 3059/6-1, NERC-1633948, NSFC-41661144004, and NSF-1633948). D.S.S. was financed through the AXA Chair for Functional Mountain Ecology, funded by the AXA Research Fund through the project GloMEc. A.L. and R.B. were funded by the BioDiversa project FishME (ANR-21-BIRE-0002-01) obtained in the call on ecosystem restoration. M.C.F. was funded by NERC and the CIFAR Fungal Kingdoms program.

R.B. contributed to formal analysis, visualization, and writing of the original draft. H.S., K.A.B., and M.C.F. contributed to data collection and reviewing and editing. D.S.S. contributed to conceptualization, funding acquisition, investigation, methodology, project administration, resources, supervision, and reviewing and editing. A.L. contributed to conceptualization, formal analysis, funding acquisition, investigation, methodology, resources, supervision, and reviewing and editing.

### AUTHOR AFFILIATIONS

[1]Toulouse INP, CNRS, IRD, CRBE, Université de Toulouse, Toulouse, France

[2]Chrono-environnement (UMR 6249), CNRS, Université Marie et Louis Pasteur, Besançon, France

[3]Faculty of Medicine and Dentistry, Blizard Institute, Queen Mary University of London, London, United Kingdom

[4]Department of Infectious Disease Epidemiology, School of Public Health, Imperial College London, London, United Kingdom

## AUTHOR ORCIDs

Rayan Bouchali  http://orcid.org/0000-0002-4946-547X
Hugo Sentenac  http://orcid.org/0000-0002-6535-1958
Kieran A. Bates  http://orcid.org/0000-0003-1559-6014
Matthew C. Fisher  http://orcid.org/0000-0002-1862-6402
Dirk S. Schmeller  http://orcid.org/0000-0002-3860-9933
Adeline Loyau  http://orcid.org/0000-0003-4230-5892

## FUNDING

| Funder | Grant(s) | Author(s) |
|---|---|---|
| Belmont Forum | ANR-15-MASC-0001-P3, DFG-SCHM 3059/6-1, NERC-1633948, NSFC-41661144004, NSF-1633948 | Dirk S. Schmeller |
| AXA Research Fund | GloMEc | Dirk S. Schmeller |
| Biodiversa+ | ANR-21-BIRE-0002-01 | Rayan Bouchali |
| | | Adeline Loyau |
| Natural Environment Research Council | | Matthew C. Fisher |
| CIFAR fungal kingdom program | | Matthew C. Fisher |
| BBSRC New Investigator Award | | Kieran A. Bates |

## AUTHOR CONTRIBUTIONS

Rayan Bouchali, Formal analysis, Visualization, Writing – original draft | Hugo Sentenac, Data curation, Writing – review and editing | Kieran A. Bates, Writing – review and editing | Matthew C. Fisher, Writing – review and editing | Dirk S. Schmeller, Conceptualization, Funding acquisition, Investigation, Methodology, Project administration, Resources, Supervision, Writing – review and editing | Adeline Loyau, Conceptualization, Formal analysis, Funding acquisition, Investigation, Methodology, Resources, Supervision, Writing – review and editing

## DATA AVAILABILITY

The data sets supporting the conclusions of this article are available in NCBI/ENA/DDBJ under the accession number PRJEB91889.

## ETHICS APPROVAL

Fieldwork was carried out with multi-year permit nos. 2016-110, 2016-111, and 2017-s33 delivered by the Parc National des Pyrénées and the Région Occitanie.

## ADDITIONAL FILES

The following material is available online.

## Supplemental Material

**Supplemental Material (mSystems01436-25-s0001.docx).** Figures S1 to S9; Tables S1 to S12.

## Open Peer Review

**PEER REVIEW HISTORY (review-history.pdf).** An accounting of the reviewer comments and feedback.

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
