## [Reviewer comments · mSystems]

Unravelling the disease pyramid: the role of environmental micro-eukaryotes in amphibian resistance to the deadly fungal pathogen *Batrachochytrium dendrobatidis*

Rayan Bouchali, Hugo Sentenac, Kieran Bates, Matthew Fisher, Dirk Schmeller, and Adeline Loyau

Corresponding Author(s): Rayan Bouchali, Toulouse INP

Review Timeline:

Submission Date:

October 10, 2025

Accepted:

November 12, 2025

Editor: Angela Oliverio

Reviewer(s): The reviewers have opted to remain anonymous.

Transaction Report:

DOI: <https://doi.org/10.1128/msystems.01436-25>

Re: mSystems01436-25 (**Unravelling the disease pyramid: the role of environmental micro-eukaryotes in amphibian resistance to the deadly fungal pathogen *Batrachochytrium dendrobatidis***)

Dear Dr. Rayan Bouchali:

I am happy to share that your manuscript has been accepted. Thank you for the detailed responses, additional analyses, and thoughtful revisions. I am forwarding it to the ASM production staff for publication. Your paper will first be checked to make sure all elements meet the technical requirements. ASM staff will contact you if anything needs to be revised before copyediting and production can begin. Otherwise, you will be notified when your proofs are ready to be viewed.

Sincerely,
Angela Oliverio
Editor
mSystems